# Methodology for Designing Systems Based on Tangible User Interfaces and Gamification Techniques for Blind People

**Luis Roberto Ramos Aguiar** * and **Francisco Javier Álvarez Rodríguez** *

Departamento de Ciencias de la Computación, Universidad Autónoma de Aguascalientes, Aguascalientes 20100, Mexico

* Correspondence: roberto.ramos.nay@gmail.com (L.R.R.A.); fjalvar.uaa@gmail.com (F.J.Á.R.)

**Abstract:** Having a disability does not mean being away from major technologies present today; even people with visual impairment or blindness use different options to access technological information. Recent studies have shown that using tangible user interfaces and gamification techniques brings considerable benefits to learning and the understanding of essential topics for these people. Therefore, METUIGA methodology has been developed to facilitate digital content creation that mixes both characteristics and seeks to take advantage of the primary means of knowledge that these people have as their sense of touch, enriched with techniques that encourage them to use applications more frequently. For this reason, novelties are shown within the requirements and the design stages to implement these techniques. This work shows prototypes that have been made following METUIGA methodology to help teach geometry and mathematical lessons for blind people. In addition, a third prototype focused on children with an autism spectrum disorder demonstrates how METUIGA methodology can be applied in a variety of subjects and for a number of disabilities. Finally, an analysis of the software methodology evaluation is presented to show the initial perceptions of software developers toward METUIGA methodology, where important results were obtained in relation to the software engineering process application.

**Keywords:** software development; tangible user interfaces; gamification; blind children; computational applications

## 1. Introduction

Throughout history, disability has been considered through different points of view and although the ways of approaching it have changed radically, in each historical stage there have been supportive or discriminatory attitudes towards people with disabilities [1]. According to the World Health Organization, a disability refers to persons who have one or more impairments, whether physical, mental, intellectual, or sensory, which, due to their interaction with different environments in the social setting, may impede their full and effective participation on equal terms [2,3].

It is estimated that more than 1 billion people live with some form of disability. This figure corresponds to approximately 15% of world's population; in fact, up to 190 million (3.8%) of people aged 15 years and older have significant difficulties functioning and frequently require health care services [2]. Furthermore, technology's involvement in all areas of daily life is unquestionable, the development of these resources is growing impressively and in this context the fields of disability and pedagogy have had a great impact. Therefore, it is of great importance to know the different alternatives provided using technology, as well as to redesign the teaching–learning systems in all educational spaces.

Technologies have evolved over the years to achieve greater familiarity and interactions with users, and different methods have gradually appeared that can improve their experience with the technology presented to them. For example, recommender systems have become essential to users for finding "what they need" [4] and in virtual reality, users

can experience changes of position in space and respective perspectives that they cannot experience in the real environment [5]. We have also augmented reality, a variation of virtual reality that, in contrast to virtual reality, encourages users to observe the real world with virtual objects superimposed on or composited with the real world. Thus, augmented reality complements reality, rather than replacing it completely [6]. Likewise, we have tangible user interfaces that allow interactions with digital technology through physical environments, so that instead of looking at graphical interfaces, they are seamlessly integrated into the user's physical environment [7]. Despite these great advances in terms of interactions with digital information, those who suffer from disability remain aside. Within this vulnerable group are those who suffer from low vision problems or blindness and those who have alterations in the vision organ that affects the main medium used to consume digital information. Moreover, low vision or blindness does not isolate blind people, but drives them to consume as much information as they can acquire to develop in the same way as sighted people. The most pressing need of people with low vision and blindness, in addition to society's awareness of disability, are technological requirements. Technical and technological aids significantly favor people's access to the mass media [8]. Therefore, it is of the utmost importance to propose technological elements that focus on the main senses of exploration based on the person to whom the system is addressed.

METUIGA is a software design methodology that aims to facilitate the design of technological educational systems to help support essential subjects for blind people, and here you can find a series of activities to implement tangible user interfaces. It is a technology that allows interactions with digital information through different physical objects, thus exploiting the main sense of blind people, which is through touch. It also implements gamification techniques to provide enriched systems with challenges, missions, mechanics, and dynamics to encourage blind people to interact with the application on a frequent basis. Once the stages and activities of the methodology are defined, to demonstrate the results that have been obtained in the implementation of this methodology, three prototypes are presented. The first one is aimed at teaching geometric topics to blind children and the second one is focused on teaching mathematical portions to blind children. Both prototypes were designed with the intention of contributing to children's organizational skills, spatial orientation, and difficulty with rational numbers. The third prototype shows an experimentation of the METUIGA methodology with the creation of a software to teach emotions to children with an Autism Spectrum Disorder (ASD).

Considering that software development must comply with a quality process, it was decided to carry out an evaluation of the methodology by experts in software engineering who followed the development process by building applications oriented to blind people. Afterwards, an evaluation model for software development methodologies was applied, which has as its objective "quality assurance in the development of projects, reduction or mitigation of risks and the ease of defining project deliverables" [9].

The remainder of this paper is composed as follows: Section 2 presents the problems of developing applications for people with disabilities, Section 3 presents the background of tangible user interfaces and gamification, Section 4 presents the METUIGA methodology and gives a brief explanation about the activities performed in its stages, Section 5 shows the prototypes generated with the METUIGA methodology, Section 6 explains the model of the evaluation of the software development methodologies proposed by the authors in [9], Section 7 presents a discussion related to the data obtained in the applied evaluation model, and finally Section 8 shows the conclusions of this work.

## 2. Problem

The development of applications oriented to people with disability is a challenge not commonly faced by development teams. Because of this, they do not know the essential features to cover in building these applications. For example, applications aimed at people with visual impairments should consider the most intuitive means of interaction to present digital information. These means of interaction may change according to the capabilities

of the end user; it is not the same as designing an application for people with hearing impairments or any other disability, although the subject to teach is the same.

To help with this problem, new methodologies have emerged in software development that focus on different disabilities with aim of helping in the construction of applications aimed at these people, thus facilitating an understanding of digital information, and taking advantage of the main means of interaction according to the disability to be treated.

An important aspect within the development process of these methodologies is the integration of the users with disabilities in all the stages and activities related to building the software. This characteristic has its origins in user-centered design (UCD). To explain the UCD concept, we show some of the authors' definitions that can be found in the literature. For example, according to Domingo and Pera [10] UCD "is a way to plan, manage and carry out projects of creation, improvement and implementation of interactive products. At the same time, it is also considered a design philosophy or approach according to which any design activity must consider for whom it is designed, as well as the contexts of use". On the other hand, Abras et al. [11] defines UCD as "a general term for a philosophy and methods that focus on designing for users and their participation in the design of computer systems. The forms of user participation can vary. At one end of the spectrum, participation can be relatively light: they are consulted about their needs, observed, and participate in usability testing". If we look for a more pointed definition, we can also observe the definition of Fernandes and Healy [12] who describe UCD as "a methodological set in which it is assumed that the entire design process should be driven by the user, their needs, objectives and characteristics".

Therefore, UCD can be defined as an essential approach in all methodologies that intends to build applications aimed at some type of disability. The importance of knowing for whom it is designed, the context of the digital information to be presented, the knowledge that is expected to be achieved, the best methods of interaction to achieve greater familiarity, and the avoidance of possible frustrations when individuals feel a system is not built for them, are fundamental characteristics when building this type of application.

## 3. Background

METUIGA is the first methodology that mixes tangible user interfaces and gamification techniques within its development process. Definitions of these characteristics are shown below.

### 3.1. Tangible User Interfaces

In the last two decades, Tangible User Interfaces (TUIs) have emerged as a new interface type that interlinks the digital and physical worlds. Drawing upon users' knowledge and skills of interaction with the real non-digital world, TUIs show potential to enhance the way in which people interact with and leverage digital information [13]. These interfaces were identified by Fitzmaurice, although he called them "manipulable user interfaces" [14]. His definition was that of a physical object that performs a virtual function in which the physical object serves as a manipulator of different virtual events. Furthermore, TUIs were defined as devices that shape digital information, using physical objects as representations and controls of computational data led by the Tangible Media Group of the MIT Media Lab [15].

The term tangible refers to physical elements that will be part of tangible interfaces with which the end user interacts. Its role is to represent digital information with which activities will be performed. Tangible user interfaces can give physical form to digital information, facilitating the manipulation of different elements related to educational or bone context. Designers of TUIs are always looking for the union of the physical with the virtual. TUIs can make accessible the input of information through physical surfaces (walls, desks, ceilings, windows), environmental media (building blocks, models, instruments) and environmental media (light, sound, air flow, water flow) within physical environments. TUIs support collaboration between different users, who can interact with

the environment simultaneously, and physical models of TUIs provide the user with an intuitive understanding of complex structures. Potential users of TUIs can be people who are not always sitting in front of a computer and people who do things better with their hands and objects.

Talking specifically about tangible user interfaces for blind people, a study by Maria et al. [16] who created a system called "Touch&Learn" for teaching new Braille to blind children considered that these interfaces provide a new way to learn basic concepts. Furthermore, MICOO (Multimodal Interactive Cubes for Object Orientation) is a tangible interface for blind or visually impaired people to create, modify and interact with diagrams and graphics. Their creators consider that these interfaces reduce the need for manual intervention and allow independent discovery by the users (blind people) and provide dynamic behavior while interacting and providing feedback [17]. Likewise, the use of tangible interfaces can be extended to other areas, such as the work completed by Morrison et al. [18] whose study, "Torino: A Tangible programming languages inclusive of Children With Visual Disabilities" showed that that the use of physical objects to represent or interact with programming constructs may be a more obvious choice for children with visual disabilities. In addition, Sitdhisanguan et al. [19] in their work "Using tangible user interfaces in computer-based training systems for low-functioning autistic children" considered that in terms of learning efficiency, TUIs offer a greater improvement compared to traditional tactile training methods. Additionally, Francis et al. [20] in his work "Do Tangible User Interfaces promote social behaviour during free play? A comparison of autistic and typically-developing children playing with passive and digital construction toys" concludes that for children with ASD this type of interface may be beneficial as it has been designed with sensory aspects of embodiment in mind. Moreover, Al Mahmud and Soysa [21] considered that tangible interfaces can link real-world physical objects with the digital world, which has a lot of potential for children with ASD. Finally, Price et al. [22] in their work "Using 'tangibles' to promote novel forms of playful learning" the successful use of tangible arrangements for engendering playful learning in several ways was shown.

### *3.2. Gamification*

Gamification is a concept that emerged in educational literature at the beginning of the 21st century and it is presented from multiple definitions and perspectives. Ramirez considers it as the utilization of strategies, thoughts and game mechanics in non-habitual contexts for the modification of the participants' behavior [23]. According to Kapp, gamification is "the use of mechanics, aesthetics, and game thinking to engage people, motivate action, promote learning, and solve problems" [24]. On the other side, "gamification is the use of game thinking, approaches and game elements in a context other than games" [14].

Several definitions are intertwined with their terms about gamification, but it could be summarized as: gamification is the integration of game elements, activities, and challenges that are not necessarily games, but help keep the user hooked because of their interest in achieving the various challenges presented to them. Games use some elements that play an important role in gamified systems:

- Activities and challenges that the user performs to increase their progress towards new defined objectives;
- Points are accumulated as different activities or challenges are completed;
- The user's level increases as he/she overcomes the different defined objectives;
- Rewards or badges that show you have passed or completed an activity;
- Ranking in relation to other users to compare your progress.

With new game engines that have offered free availability to developers to build their projects, the implementation of gamification techniques has been influenced by virtual reality applications that seek to immerse users in digital environments [5,25]. To applications aimed especially at blind people that have been shown to improve interest and enthusiasm for learning and thus increase understanding of the presented topics, recent studies have shown that the inclusion of challenges and interaction with gamification

elements in mobile applications motivates them to learn but generates some confusion when identifying the correct part of the screen when they must select the correct part of the screen [26,27].

## 4. METUIGA Methodology

METUIGA (MEthodology for the design of systems based on Tangible User Interfaces and GAmification techniques for blind people) is result of analysis and research, carried out with the aim of generating a methodology focused on developing interactive educational systems following user-centered design principles and the iterative cascading life cycle, to create applications that take advantage of tangible user interfaces and gamification techniques to improve learning and understanding in blind people. Some of the important aspects of the METUIGA methodology are as follows:

- Activities defined to build a tangible interface, the classification of tangible objects and software used to track tangible objects.
- It defines a process composed of five activities to define the gamification characteristics that the system will have to engage the target users.
- It defines different evaluation activities to measure usability in blind people and to measure the fulfillment of the proposed objectives.
- It defines a process focused on user-centered design to generate systems that adapt to the needs of end users.

METUIGA methodology is composed of four main stages (Requirements, Design, Implementation, Evaluation) nurtured with activities related to tangible user interfaces and gamification. Within them different prototypes can be made where the end user is included in each of them to achieve products closer to their needs. Below, Figure 1 shows the methodological process of the METUIGA methodology where the horizontal arrow shows the stages of requirements, design, implementation, and evaluation manifesting the possibility of interactions between each of them in a flexible manner. Moreover, vertical arrows indicate an interaction between each stage with the end user:

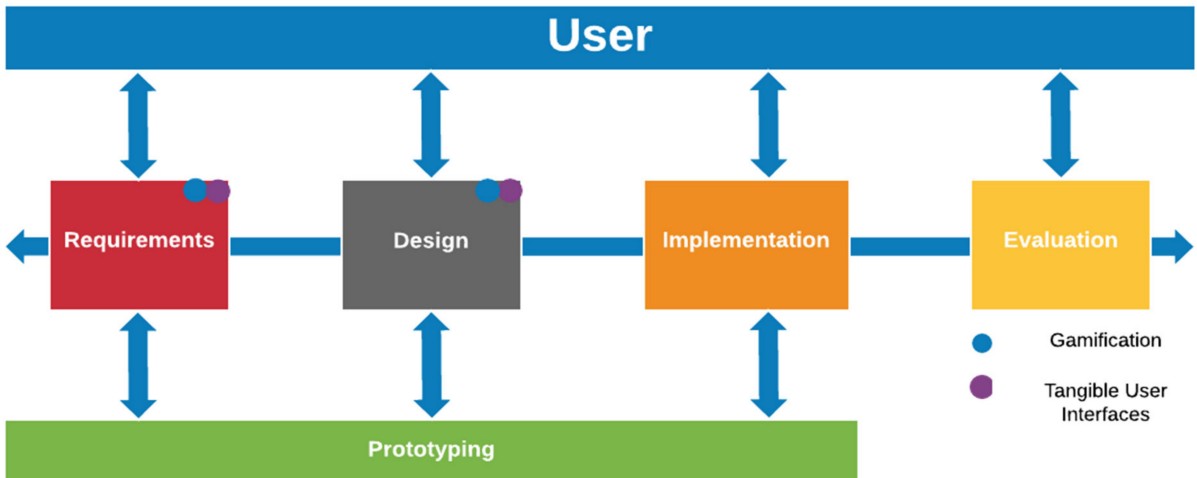

**Figure 1.** METUIGA methodology.

To compare the novelties within the METUIGA methodology, an analysis of different user-centered software development methodologies was carried out, to analyze if they included any disability, if gamification techniques and tangible interfaces were used within their development process, and if the disability type was addressed to blind people. Table 1 shows the analyzed methodologies and the evaluated criteria:

**Table 1.** Software methodologies analysis.

| Software Methodology | Disability | Gamification | TUI | Blind People |
|---|---|---|---|---|
| Mpiu + a [28]. | NO | NO | NO | NO |
| MICEE [29]. | YES | NO | NO | NO |
| MECONESIS [30]. | YES | YES | NO | NO |
| MPOBA [31]. | NO | NO | NO | NO |
| METUIGA. | YES | YES | YES | YES |

Below, you can see the steps that make up each of the METUIGA methodology stages, whose objective is to help build applications that implement tangible user interfaces and gamification techniques for blind people.

### 4.1. Requirements Stage

The requirements stage aims to define the needs of the system, the tangible interfaces, and the gamification. This stage is composed of five activities that allow us to first establish the characteristics of the software engineering, the gamification, and the tangible user interfaces for the future system to be developed. In relation to the gamification characteristics, we identified the most appropriate type of gamification to apply in the future system, to be developed with the objective of establishing mechanics and dynamics according to the target user in the design phase. For this we resorted to the types of gamification proposed by the authors in [32] (See Table 2). Likewise, one of the most important challenges when trying to develop applications that implement tangible user interfaces is identifying the necessary tools to create them. For this reason, to help accelerate this process, METUIGA methodology provides a tool where you can observe the existing works aimed at blind, visually impaired people, or people in general to identify the most appropriate tools according to the needs of the project (See Table 3).

**Table 2.** Categories or types of gamifications in requirements stage. Source: [32].

| Type of Gamification | Description |
|---|---|
| Internal | To improve motivation within an organization |
| External | When seeking to involve customers by improving the relationship between them and the company. |
| Behavioral change | It seeks to generate new habits in the population, from encouraging them to choose healthier options to redesigning the classroom and encouraging them to learn more while enjoying themselves. |

**Table 3.** Suggested projects to identify tools using tangible user interfaces in requirements stage.

| Name | Description |
|---|---|
| Tangible User Interfaces to Ease the Learning Process of Visually-Impaired Children [33]. | NFC technology is used to enable users to interact with physical objects using the tangible interface. Objects are identified with NFC tags placed inside them transparently to the user. The application is deployed on a mobile device with an NFC reader to recognize the objects and react accordingly. |

**Table 3.** *Cont.*

| Name | Description |
|---|---|
| MICOO (Multimodal Interactive Cubes for Object Orientation): A tangible user interface for the blind and visually impaired [17]. | MICOO has been developed from scratch, using affordable and readily available hardware components. The system enables natural interaction using MICOO on a multi-touch surface. The tangible interface is controlled via multiple infrared cameras to track MICOO's movement. |
| A tangible user interface-based application utilizing 3D-printed manipulatives for teaching tactual shape perception and spatial awareness sub-concepts to visually impaired children [16]. | For object tracking at tangible interface, they use Trackmate, an open-source tangible tracking system based on computer vision, introduced by the Tangible Media Group at the MIT Medioa Lab. |
| Using tangible user interfaces in computer-based training systems for low-functioning autistic children [19]. | A web camera placed underneath the tabletop to detect and identify the chosen tangible object. Infrared LED tubes were embedded on each tangible object. Each wood block or toy contains a unique number of tubes. Thus, from the picture taken by a web camera, the number of bright spots can be used to identify the selected and placed object. |
| Designing tangibles for children: What designers need to know [34]. | Introduces four areas of cognitive development which may be relevant for the design of tangible systems. |
| Using Developmental Theories to Inform the Design of Technology for Children [35]. | Concerns the development of a new educational resource for early childhood technology education. The development has involved a detailed analysis of the target audience, the formulation of appropriate design criteria, the construction of the resource and its subsequent evaluation. |
| Evaluating children performance with graphical and tangible robot programming tools [36]. | Presented a comparison study of children's performance using the two isomorphic subsystems (TUI vs. GUI). |

Figure 2 shows the stages that make up the requirements stage:

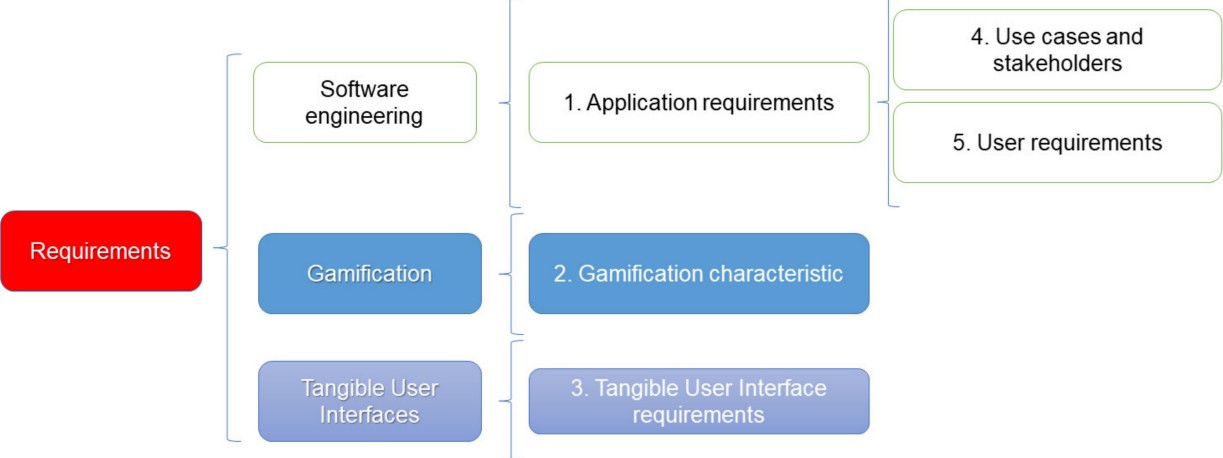

**Figure 2.** METUIGA methodology requirements stages.

*4.2. Design Stage*

The design stage consists of defining the graphical interfaces of the system considering the gamification features and the tangible user interfaces. This stage is composed of twelve activities, with which the system established in the requirements stage has designed in relation to the gamification techniques together with the integration of tangible user interfaces to the developed system. In the gamification part, we found the identification of challenges and missions to be implemented, the classification of missions into common, individual, and collective, the design of a gamification structure, the selection of mechanics and dynamics to be implemented in the gamified system, and the design of an avatar interface (See Table 4).

**Table 4.** Gamification activities to be followed within METUIGA methodology in design stage.

| Type of Gamification | Description |
|---|---|
| Challenge and misión | Within a gamified system it is important to include challenges and missions to keep the users' attention. Strong fun is associated with challenges, based on the pleasure of overcoming the challenge [37,38]. |
| Mission classification | A mission is composed of one or more challenges, in this section we classify the challenges that will be part of each mission. |
| Gamification structure | Following a gamification structure is important to achieve a good end-user satisfaction. For this reason, a structure composed of six components (Challenges, Conditional Wins, Leaderboards, Medals, Social Networks, Status) is provided to choose the most appropriate structure to develop the project. |
| Choice of game mechanis and game dynamics | Mechanical techniques are ways of rewarding users according to objectives achieved (accumulation of points, level status, obtaining prizes, rewards, rankings, challenges, missions or challenges), while dynamic techniques refer to users' own motivation to play and move forward in achieving their objectives (rewards, status, achievement, compensation) [39]. |
| System avatar | Personalization is an important aspect in gamification implementation, for this reason an avatar creation is proposed in case that development team considers it appropriate to encourage interaction between normal-visual people and blind people. Even a simple player headshot and screen name can be considered an avatar, providing player with an opportunity for customization [40]. |

After defining the gamification features to be implemented, we proceeded to the software engineering stages, such as the design of class diagrams, flow charts, construction of static and dynamic designs of system interfaces, construction of a database and finally, we classified and designed the features of tangible objects that will interact within the developed system, establishing their behavior when interacting with tangible interfaces and the user. For this purpose, we used the TAC palette that allows developers to visually describe the behavior of tangible objects when in contact with the tangible interface, thus linking the graphical interfaces developed with the gamified elements with the tangible objects developed (see Figure 3).

| TAC | Representation | | | Association | Manipulation | |
|---|---|---|---|---|---|---|
| | Variable | Type | Constraint | Associated graphic | Action | Response |
| 1 | Construction | Construction model | Surface activated can interac with other constructions | | Add | Display the shadow of the object according to the time |
| | | | | | Remove | Remove information displayed on the screen |
| | | | | | Move | Update its position on the screen, Display its posicion on the screen. |
| 2 | Distance | Distance tool | Must be two construction objects on the surface | | Add | Distance between objects is displayed |
| | | | | | Remove | Hide distance of objects |
| 3 | Simulate wind | Wind shearing | At least one construcción object must be present on the surface | | Add | Show wind on screen |
| | | | | | Remove | Hide wind show non screen |
| | | | | | Move | Update your position on the screen |

**Figure 3.** TAC palette to describe behavior of tangible objects when interacting with a tangible interface. Source: [41].

Once the tangible object behaviors have been defined, it is necessary to build the tangible interfaces. To complete this, it is important to note that during the requirements stage, the development team must have already decided which tool best suits the needs of the project. In case of the team not having decided yet which tool to use, METUIGA methodology in its twelfth activity of the design stage recommends that Reactivision [42] and its diagram is used to build a tangible interface (See Figure 4).

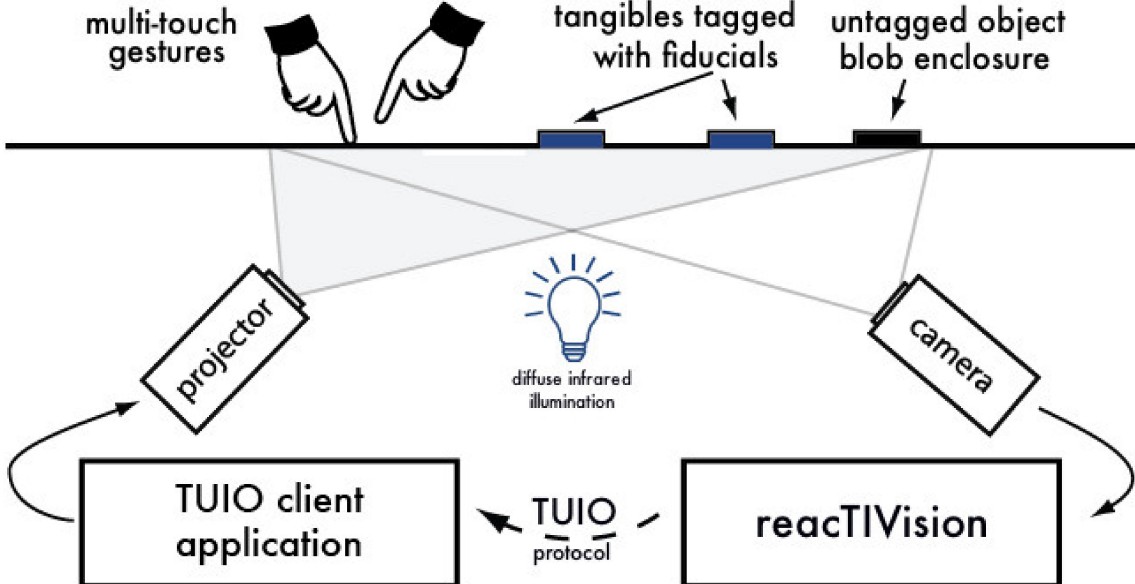

**Figure 4.** Diagram to build a tangible interface proposed by Reactivision. Source: [42].

Figure 5 shows the stages that make up the design stage, where the integration of gamification elements, software engineering and tangible user interfaces make the

METUIGA methodology the first to establish a design process that considers these three characteristics. See Appendix A for the activities involved at this stage.

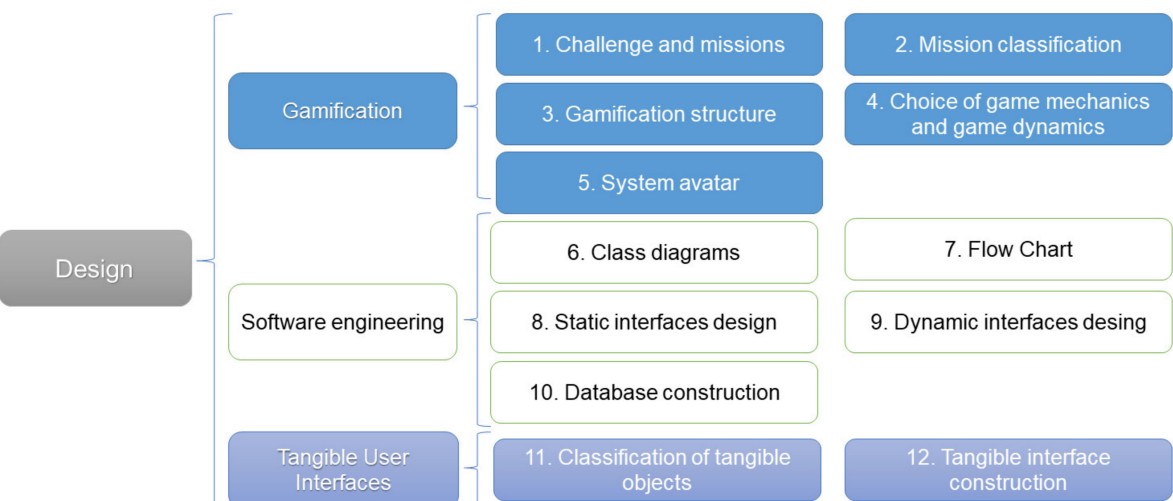

**Figure 5.** METUIGA methodology design stages.

### 4.3. Implementation Stage

The implementation stage consists of programming what was established in the design stage. This stage consists of three activities starting with the identification of a programming standard, the programming of the software based on previously created designs and finally, the design of a developed system architecture. Figure 6 shows the stages that make up the implementation stage:

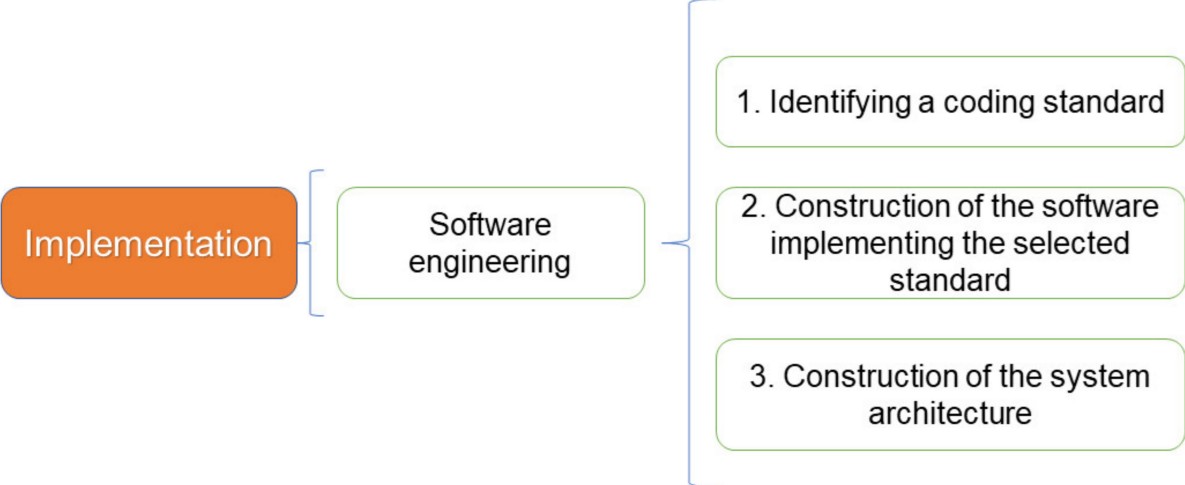

**Figure 6.** METUIGA methodology implementation stages.

### 4.4. Evaluation Stage

The purpose of the evaluation stage is to determine the acceptance of the system by end users and the fulfillment of the objectives established in the requirements stage. This stage is composed of three activities, starting with a measurement of the usability of the system using a "System Usability scale" instrument [43], followed by an evaluation of the system objectives by means of a "Light MECPDS" instrument [44] and finally, a measurement of the usability through a "Thinking Out Loud" instrument [45]. This specific instrument helps to measure usability when making applications for blind people. Figure 7 shows the stages that make up the evaluation stage:

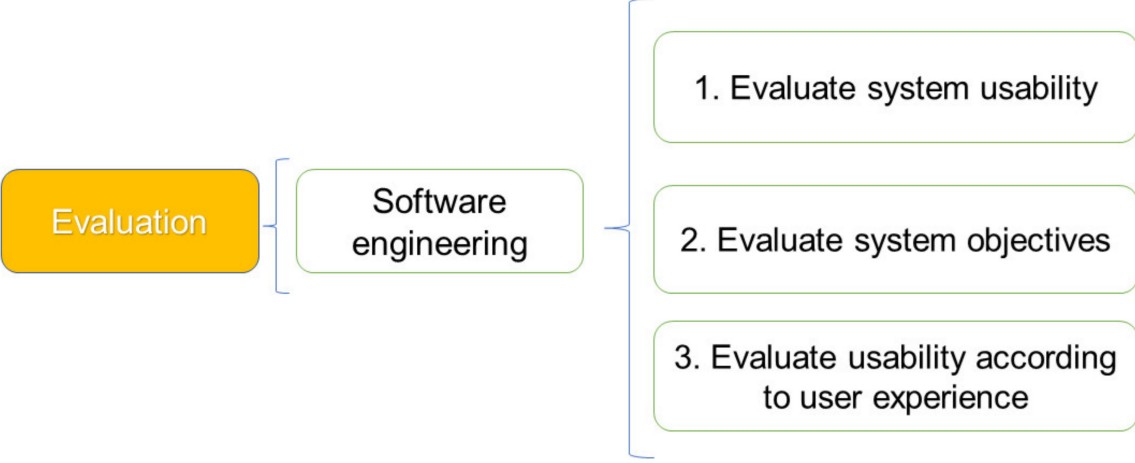

**Figure 7.** METUIGA methodology evaluation stages.

It is important to mention that although these instruments are proposed by the METU-IGA methodology, the development team is free to use other types of tools to evaluate the developed systems. For example, Zaman et al. [46] in their work "Editorial: the evolving field of tangible interaction for children: the challenge of empirical validation" presented empirical articles reporting on studies that obtained complementary knowledge about children's perceptions of tangible interaction. Moreover, the benefits of implementing tangible interfaces are detailed in the paper "Are Tangibles More Fun? Comparing Children's Enjoyment and Engagement Using Physical, Graphical and Tangible User Interfaces" by Xie et al. [47] who present a qualitative analysis based on observational notes and audio responses to open interview questions to help contextualize the quantitative findings and provided key insights into interactional differences not apparent in the quantitative findings. Likewise, you can also analyze the work of Sapounidis et al. [48], "Latent Class Modeling of Children's Preference Profiles on Tangible and Graphical Robot Programming" where an instrument that used eight variables to operationalize children's preferences were observed: attractiveness, collaboration and usability factors, such as enjoyment, ease of use, convenience, understandability, explainability and ease of recall. Futhermore, in a paper titled "Tangible and Graphical Programming with Experienced Children: A Mixed Methods Analysis" you can observe a study that applied a mixed methods approach to analyze the usability, collaboration, and playfulness aspects in introductory programming activities with tangible and graphical user interfaces of two groups of students [49].

## 5. Developed Applications

Building a TUI is a complex process that encompasses multidisciplinary knowledge including engineering, art, and social sciences [50]. Therefore, to build prototypes focused on teaching educational topics aimed at blind children, cooperation between different areas of knowledge was necessary (see Figure 8). After defining the stages of the METUIGA methodology, educational experts who had previously worked with blind people were consulted to choose the educational topics that, in their view, with tangible user interfaces could enhance their learning and understanding using tangible objects. Likewise, due to the UCD approach followed by METUIGA methodology, tangible interfaces were designed in close collaboration with educators and blind people. One of the most important benefits of this multidisciplinary approach is that designers can move beyond the focus on computers to physicality, namely from screens and applications on smartphones to tangible interfaces [51,52]. Subsequently, programmers oversaw reflecting what was established by the designers.

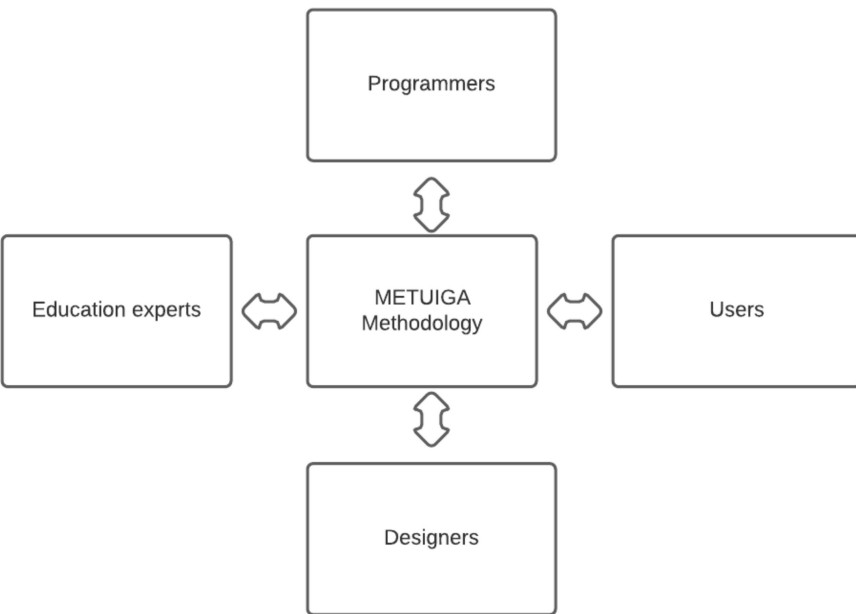

**Figure 8.** Multidisciplinary areas working together to create prototypes.

The developed prototypes demonstrate that it is possible to generate applications that combine the characteristics of tangible user interfaces and gamification techniques to teach educational topics to blind people, providing a new interaction experience. This new experience focuses on providing a computer application enriched by gamification elements with which the user interacts through physical elements. On the other side, due to the lack of vision of the target users, we resorted to auditory feedback to indicate the different states of the gamified elements (see Figure 9).

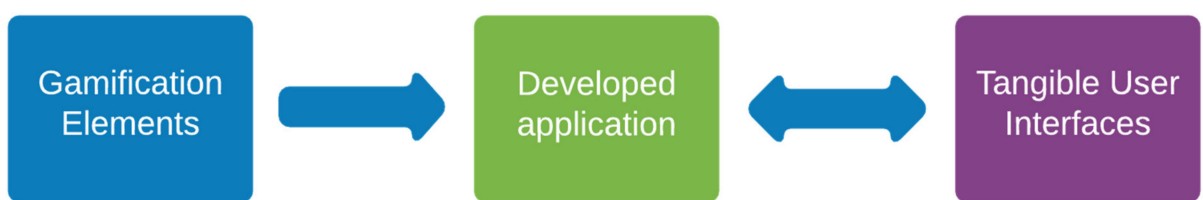

**Figure 9.** Interaction between the gamification elements, the developed application, and the tangible user interfaces.

### 5.1. Prototype One

According to Nuñez "geometry contributes strongly to development of organization and spatial orientation skills of the blind student" [53]. The first prototype developed is called GeoTang (Tangible Geometry), which aims to help blind elementary school children learn basic geometry concepts by recognizing geometric objects (triangle, rectangle, square, circle, pentagon, hexagon, rhombus, oval, rectangle) that are available in the application and have a tangible object to identify them.

The child interacts with the application using the tangible interface and receives auditory feedback. The application has two options. In the first one, objects are placed on acrylic, and they receive object audio feedback in the question. In the second option, the questions are asked in form of riddles to be answered by placing the correct figure on the acrylic of the tangible interface.

The system is enriched with a series of gamification elements such as a system of lives, challenges, mission, avatars, levels, coins, and rewards that are obtained according to performance. Due to the lack of visibility of the target users, sounds were added to indicate when a correct or incorrect answer is given, as well as to indicate gamification mechanics and dynamics suggested by the methodology. On the other hand, special rewards were

provided for the use of different tangible objects. Likewise, one of the important aspects in developing this application was to promote the interaction between normal-visual and visually impaired or blind users. For this reason, it was decided to implement the delivery of virtual coins to complete the challenges posed in the application so that they can be exchanged in a virtual store for different avatars and thus promote interaction between both users. Moreover, some of these challenges and missions were related to different tangible objects to encourage exploration of these. Moreover, a specific audio was assigned at the time of completing a challenge to indicate that you have achieved an achievement. In addition, a voice describes the challenge or mission achieved. In this way, we managed to provide a better interaction with the system and the achievements to the blind children (see Table 5).

**Table 5.** Some challenges and integrated missions to encourage tangible object use in GeoTang application.

| Type | Description | Rewards |
|---|---|---|
| Challenges | Uses object representing a triangle | 50 coins and a medal |
| Challenges | Uses object representing a rectangle | 50 coins and a medal |
| Mission | Uses all tangible objects | 150 coins and a medal |
| Challenges | Discover which object represents a circle | 100 coins and a medal |
| Mission | Relates 50% of questions correctly with their tangible object. | 200 coins and a medal |

Figure 10a shows the graphical interfaces of the Geotag application showing the gamification elements as well as the placement of a tangible object on the tangible interface (see Figure 10b).

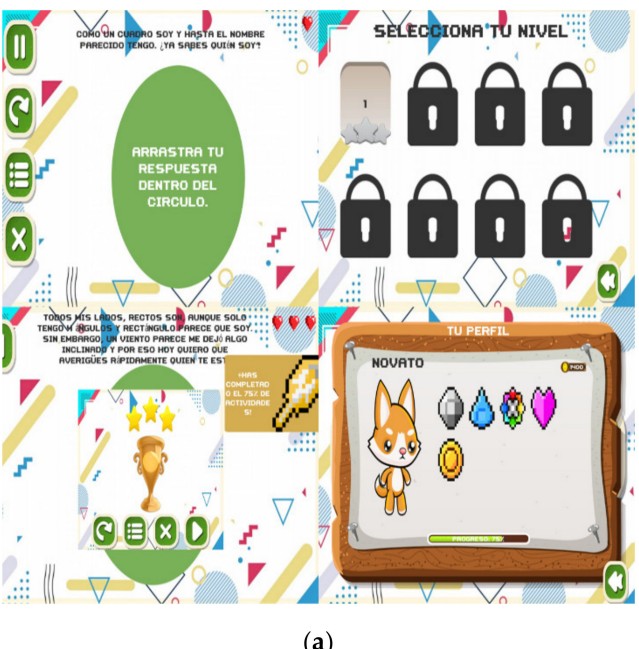

(**a**)

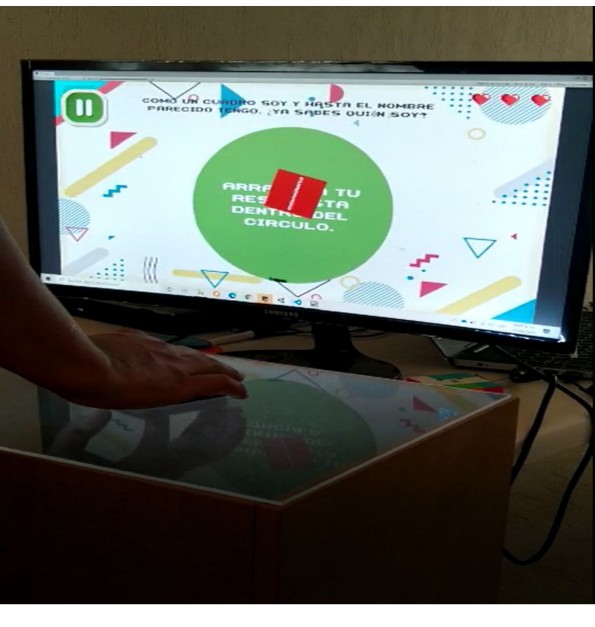

(**b**)

**Figure 10.** (**a**) Graphical interfaces of the Geotag application; (**b**) Interaction with the tangible interface by overlaying an element. Source: [54].

### 5.2. Prototype Two

The second prototype is called a tangible portion which aims to help in the learning of basic mathematical portions (an integer, a half, a quarter), due to the difficulty of understanding related to rational numbers. This difficulty is mainly because it is a subject that relies mainly on visual elements for their understanding.

The interaction with the system is quite simple: the application asks a question regarding a mathematical portion, then the child must take the object that represents the fraction that he considers to be the one we are referring to and place it on the acrylic of the tangible interface. Once the object is placed, the system recognizes it and evaluates if it is the correct answer. After this the evaluation feedback is given via audio related to their performance. Compared to prototype one, this prototype allows interaction with one or more tangible objects to represent multiple mathematical portions. For example, if a child wants to represent a 2/3 quantity, he can superimpose two objects representing 1/3 and automatically the system will detect the portion to which he wants to refer as shown in Figure 11b. The system is enriched with a series of gamification elements such as a system of lives, levels, challenges, missions, and rewards that are obtained according to performance. In comparison with prototype one, this provides less gamification features because it is specifically oriented to work with blind people. It must be considered that because of this reason, the gamification elements that are part of this application should be easily adapted to give feedback on them through audio. Likewise, some of these challenges and missions were related to different tangible objects to encourage an exploration of these. As well as prototype one, moreover, a specific audio was assigned at the time of achieving a challenge or challenge to indicate that you have achieved an achievement. In addition, a voice describes the challenge, or the mission achieved. In this way we managed to provide a better interaction with the system and the achievements to the blind children (See Table 6).

**Table 6.** Some integrated challenges and missions to encourage tangible objects use in Tangible portions application.

| Type | Description | Rewards |
|---|---|---|
| Challenges | Uses an object to represent 1/3 | Medal |
| Challenges | Uses an object to represent 1/4 | Medal |
| Mission | Uses multiple objects to represent one fraction | Medal |
| Challenges | Uses all tangible objects | Medal |

Figure 11a shows the graphical interfaces of the tangible portion's application showing the gamification elements and Figure 11b shows the multiple tangible objects interacting with the interface and the application, and you can also observe the quality system:

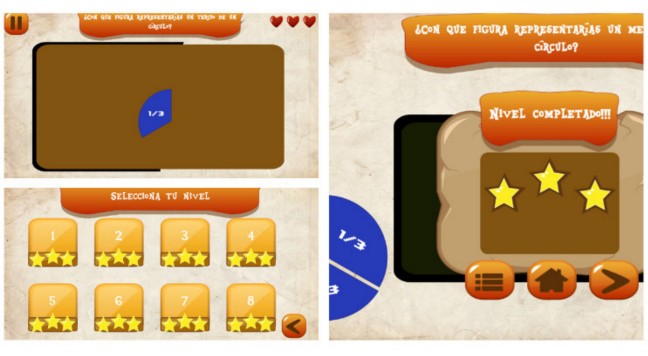

(**a**)

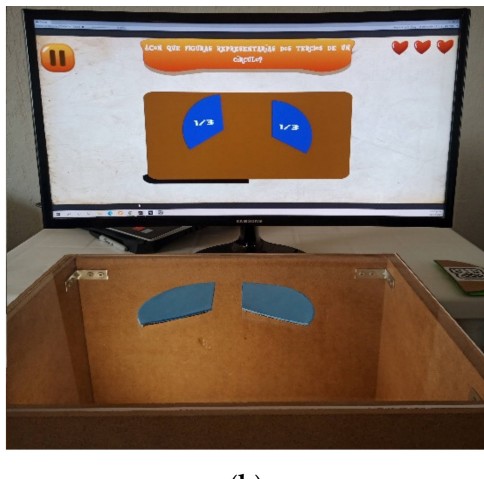

(**b**)

**Figure 11.** (**a**) Graphical interfaces of tangible portion's application; (**b**) Interaction with tangible interface by superimposing two elements to represent a composite portion. Source: [55].

### 5.3. Prototype Three

With the intention of testing the development process of the METUIGA methodology for another type of disability, it was decided to conduct an experiment to demonstrate that the generated software meets the characteristics of tangible user interfaces and gamification techniques in an application aimed at another type of target user; in this case, in people with Autism Spectrum Disorder (ASD).

"People with ASD have serious difficulties in communication and social interaction skills and behavioral flexibility" [56]. "These difficulties, which respond to variable expressions of severity, are related to the existence of barriers to learning mental states that hinder the understanding of reality from the perspective of another person. Therefore, it is important to develop teaching aimed at helping these people to acquire the relevant skills and abilities to achieve these competencies to provide them with the tools to enjoy a life as normal as possible" [57].

The third prototype is called Tangible Emotions, this application is designed to help in the recognition of basic emotions (surprise, anger, fear, sadness, happiness) for children with ASD. The application is composed of two main activities, the first activity consists of showing the main characteristics of each emotion. In the second activity the child receives a question related to a social event and with the support of an adult must deduce what emotion he/she would feel when that event occurs, once the emotion is decided the child must place the object that corresponds to that emotion on acrylic of tangible interface, then the system will evaluate his/her answer and gives the corresponding feedback.

The system is enriched with a series of gamification elements such as challenges, levels and rewards that are obtained according to performance. Furthermore, since this application is aimed at children with ASD, it was decided not to implement the system of lives so as not to generate a certain level of frustration. For this reason, it is possible to use as many tangible objects as necessary to answer each of the activities correctly.

Figure 12a shows the graphical interfaces of the tangible emotion's application showing the gamification elements and Figure 12b shows the tangible elements available for this application as well as a fiducial marker attached to one of them:

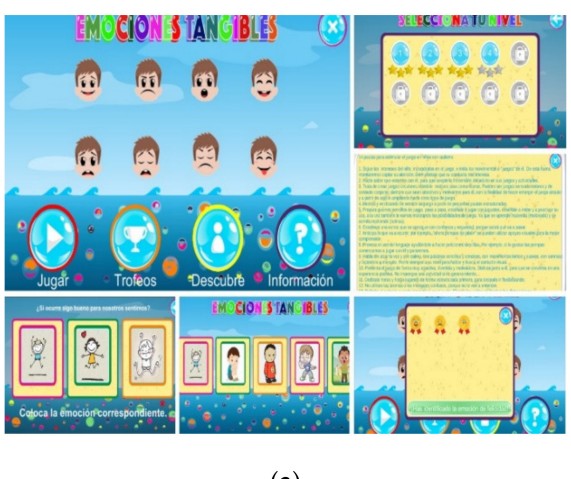

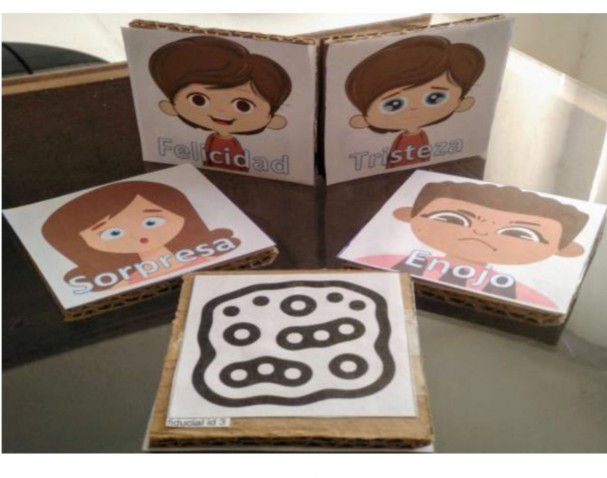

(**a**)          (**b**)

**Figure 12.** (**a**) Graphical interfaces of Tangible Emotions application; (**b**) Tangible elements of tangible emotions application and their fiducial marker.

### 5.4. Common Characteristics

All prototypes were created in Unity 3D, a multiplatform video game engine [58] working under C# language. The tangible interface is composed of a 30 cm × 40 cm acrylic, a TedGem HD camera and a 5v led light bar mounted on a 40 cm × 30 cm × 80 cm wooden structure. Moreover, an open-source tracking system for fiducial markers called ReactiVision [42] was used to track the tangible objects. Figure 13a shows the internal

components of tangible interface as well as the tangible emotions application running on the tangible interface (See Figure 13b).

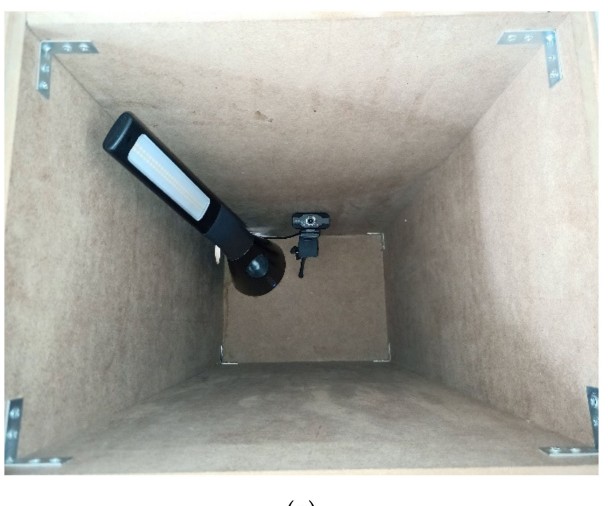

(**a**)

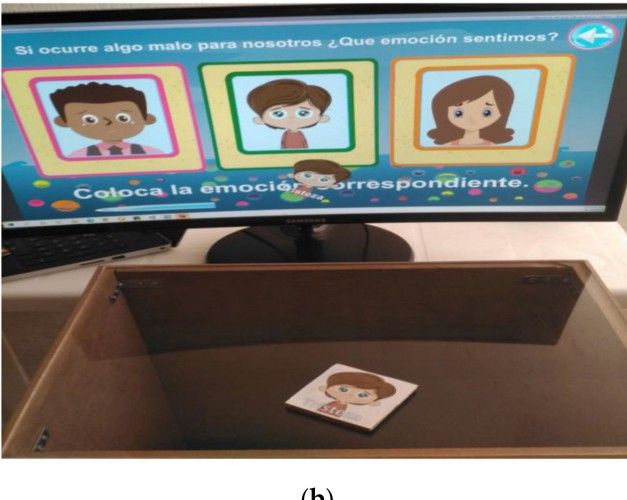

(**b**)

**Figure 13.** (**a**) Internal components of tangible interface; (**b**) Tangible emotions application running on tangible interface.

Due to the different characteristics that were desired to be implemented in developing applications, these had small variations in relation to different elements of gamification used, Table 7 shows a comparison between different elements used in applications presented here.

**Table 7.** Comparison of the gamification elements and the tangible features between the prototypes presented.

| Application | Lives | Rewards | Avatar | Levels | Progress | Multi-Objects | Feedback | Store | Virtual Money |
|---|---|---|---|---|---|---|---|---|---|
| Geotag application | * | * | * | * | * | | * | * | * |
| Tangible portion's | * | * | | * | * | * | * | | |
| Tangible emotions | | * | | * | * | | * | | |

\* This scale was weighted to obtain results where 5 was 100%, 4 was 80%, 3 was 60%, 2 was 40% and 1 was 10%.

## 6. Evaluation of Software Development Methodologies

When a new software methodology is created, it is important to know the perceptions of the software engineering experts. For this reason here is a pre-analysis of the METUIGA methodology using an instrument called an "Evaluation model of methodologies for software development". This model seeks quality assurance in project development, risk reduction or mitigation, and ease of defining project deliverables. The model proposed by Méndez and Garrido [9] consists of three stages (See Figure 14).

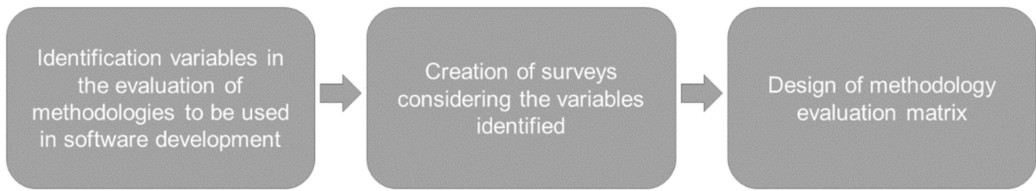

**Figure 14.** Evaluation model of a methodology. Source: Own creation, based on Méndez and Garrido [9].

### 6.1. Identification Variables in the Evaluation of Methodologies to Be Udes in Software Development

The first stage consists of defining those variables that they consider the software development methodology evaluated must comply with.

### 6.2. Creation of Surveys Considering Variables Identified

After choosing the evaluation variables, a survey is carried out with those involved using the methodology that will allow to order or prioritize each one of them, giving it a weighting according to the importance that the different people involved in software development give to each one of these characteristics. The survey is designed so that people can select the degree of importance they give to each of the variables through the following scale: 1. Strongly disagree, 2. Disagree, 3. Undecided/neutral, 4. Agree, 5. Strongly agree.

### 6.3. Desing of Software Methodology Evaluation Matrix

Finally, the methodology evaluation matrix is made, the steps to follow for the use of the matrix proposed by [9] are the following:

1. Identify key success factors;
2. Assign weighting to each key success factor;
3. Assign strength or weakness to each factor per competitor (Severe weakness = 1, Minor weakness = 2, Minor strength = 3, Major strength = 4);
4. Calculate weighted score;
5. Add results.

According to steps described previously, information needed to complete the competitive profile matrix is a tool that helps companies evaluate themselves against their main competitors using critical success factors for that industry [59]. These matrices are composed of four key components, success factors, weighting, rating and score as shown in Table 8.

**Table 8.** Competitive profile matrix.

| Success Factors | Weighting | Stage # | | Stage # | |
|---|---|---|---|---|---|
| | | Score | Weighted Result | Score | Weighted Result |
| Total score: | | | | | |

According to the results obtained from an application model of Méndez and Garrido [9] in several software development methodologies the three most important factors that must exist in a methodology are:

- Methodology must integrate the different phases of the development cycle;
- Methodology must cover the entire software development cycle;
- Methodology must be able to be used in a wide software project environment.

## 7. Results

The results obtained from Méndez and Garrido's [9] model to evaluate the METUIGA methodology are shown below.

### 7.1. Identification of Success Factors

The first activity, as specified on the model proposed by Méndez and Garrido [9], consists of identifying the variables that should be fulfilled by methodology. For this purpose, it is decided that the same 13 variables that they present in their work will be implemented, which are as follows: 1. Methodology must be adjusted to the objectives, 2. Methodology should cover the entire software development cycle, 3. Methodology must integrate the different phases of development cycle, 4. Methodology should include validation of the following, 5. Methodology must support the determination of system accuracy throughout the development cycle, 6. Methodology should be the basis for effective communication, 7. Methodology should work in a dynamic user-oriented environment, 8. Methodology should be specified in a broad software project environment, 9. Methodology must be teachable, 11. Methodology must be supported by CASE tools, 12. Methodology must

support the eventual evolution of system, 13. Methodology must contain activities leading to improve the software development process.

### 7.2. Weighing Success Factors with Information Technology Experts

To weigh the success factors, we selected nine experts in information technology (IT) with extensive knowledge in software engineering and who had previously worked with METUIGA methodology. Once selected, a survey was sent to them to weigh from 1 to 5 each of desirable requirements according to their importance within software engineering process, this survey was designed in such a way that they could select degree of importance of each one of the success factors through the next scale: 1 = Strongly Disagree, 2 = Disagree, 3 = Undecided/Neutral, 4 = Agree, 5 = Strongly Agree. Then a table was filled in to calculate the variable weights in a spreadsheet. The scale was weighted so that the results could be obtained, where 5 was 100%, 4 was 80%, 3 was 60%, 2 was 40% and 1 was 10% as shown in Table 9.

**Table 9.** Table to calculate variable weightings. Source: Own creation, based on Méndez and Garrido [9].

| Success Factors | Achieved Score | | | | | Weighted [1] | | | | | Total |
|---|---|---|---|---|---|---|---|---|---|---|---|
| | **1** | **2** | **3** | **4** | **5** | **0.2** | **0.4** | **0.6** | **0.8** | **1** | |
| | | | | | | **1** | **2** | **3** | **4** | **5** | |
| Methodology must be adjusted to the objectives | 0 | 0 | 1 | 4 | 4 | 0.000 | 0.000 | 0.111 | 0.444 | 0.444 | 0.867 |
| Methodology should cover the entire software development cycle | 0 | 0 | 1 | 2 | 6 | 0.000 | 0.000 | 0.111 | 0.222 | 0.667 | 0.911 |
| Methodology must integrate the different phases of development cycle | 0 | 0 | 0 | 5 | 4 | 0.000 | 0.000 | 0.000 | 0.556 | 0.444 | 0.889 |
| Methodology should include validation of the following | 0 | 0 | 2 | 4 | 3 | 0.000 | 0.000 | 0.222 | 0.444 | 0.333 | 0.822 |
| Methodology must support the determination of system accuracy throughout the development cycle | 0 | 0 | 3 | 4 | 2 | 0.000 | 0.000 | 0.333 | 0.444 | 0.222 | 0.778 |
| Methodology must be the basis for effective communication | 0 | 1 | 1 | 3 | 4 | 0.000 | 0.111 | 0.111 | 0.333 | 0.444 | 0.822 |
| Methodology must work in a dynamic user-oriented environment | 0 | 2 | 2 | 1 | 4 | 0.000 | 0.222 | 0.222 | 0.111 | 0.444 | 0.756 |
| Methodology should clearly specify who is responsible for results | 0 | 1 | 0 | 4 | 4 | 0.000 | 0.111 | 0.000 | 0.444 | 0.444 | 0.844 |
| Methodology must be able to be used in a wide software project environment. | 0 | 0 | 0 | 5 | 4 | 0.000 | 0.000 | 0.000 | 0.556 | 0.444 | 0.889 |
| Methodology should be teachable | 0 | 0 | 1 | 5 | 3 | 0.000 | 0.000 | 0.111 | 0.556 | 0.333 | 0.844 |
| Methodology must be supported by CASE tools | 0 | 0 | 3 | 5 | 1 | 0.000 | 0.000 | 0.333 | 0.556 | 0.111 | 0.756 |
| Methodology must support the eventual evolution of system | 0 | 1 | 1 | 4 | 3 | 0.000 | 0.111 | 0.111 | 0.444 | 0.333 | 0.800 |
| Methodology must contain activities leading to improve the software development process | 0 | 0 | 3 | 3 | 3 | 0.000 | 0.000 | 0.333 | 0.333 | 0.333 | 0.800 |

[1] This scale was weighted to obtain results where 5 was 100%, 4 was 80%, 3 was 60%, 2 was 40% and 1 was 10%.

### 7.3. METUIGA Methodology Evaluation Matrix

Once the weighted results were obtained, an evaluation matrix was designed using a competitive profile matrix used in strategic planning as a basis. Subsequently, the success factors of the METUIGA methodology were evaluated in two stages by IT experts who were part of the success factor weightings and who had previously worked with METUIGA methodology. These scores were multiplied by the obtained weights to obtain the results of METUIGA methodology as shown in Table 10.

**Table 10.** METUIGA methodology evaluation matrix.

| Success Factors | Weighting | Stage 1 | | Stage 2 | |
|---|---|---|---|---|---|
| | | Score | Weighted Result | Score | Weighted Result |
| Methodology must be adjusted to the objectives | 0.082 | 2 | 0.163 | 3 | 0.245 |
| Methodology should cover the entire software development cycle | 0.086 | 3 | 0.258 | 4 | 0.344 |
| Methodology must integrate the different phases of development cycle | 0.084 | 3 | 0.252 | 4 | 0.336 |
| Methodology should include validation of the following | 0.077 | 2 | 0.154 | 3 | 0.232 |
| Methodology must support the determination of system accuracy throughout the development cycle | 0.073 | 3 | 0.218 | 4 | 0.291 |
| Methodology must be the basis for effective communication | 0.077 | 3 | 0.232 | 4 | 0.309 |
| Methodology must work in a dynamic user-oriented environment | 0.071 | 4 | 0.282 | 3 | 0.212 |
| Methodology should clearly specify who is responsible for results | 0.079 | 3 | 0.238 | 4 | 0.318 |
| Methodology must be able to be used in a wide software project environment. | 0.084 | 4 | 0.336 | 4 | 0.336 |
| Methodology should be teachable | 0.079 | 4 | 0.318 | 4 | 0.318 |
| Methodology must be supported by CASE tools | 0.071 | 3 | 0.212 | 4 | 0.282 |
| Methodology must support the eventual evolution of the system | 0.075 | 4 | 0.3 | 4 | 0.300 |
| Methodology should contain activities leading to improve the software development process | 0.075 | 4 | 0.3 | 4 | 0.300 |
| Total score: | | | 3.263 | | 3.822 |

## 8. Discussion

Analyzing the results obtained in relation to the desirable requirements for a software methodology, it can be observed that for respondents it is important that a methodology covers all the software development cycle, as well as integrates different phases of development process. They also consider that it is necessary that it can be used in different software project environments, that it fits objectives, that it specifies who is responsible for the results and finally that it is easy to teach. Table 11 shows six success factors that obtained higher weights.

**Table 11.** Top six success factors obtained in the surveys.

| Success Factor | Score |
|---|---|
| Methodology should cover the entire software development cycle | 0.911 |
| Methodology must integrate the different phases of the development cycle | 0.889 |
| Methodology must be able to be used in a wide software project environment | 0.889 |
| Methodology must be adjusted to the objectives | 0.867 |
| Methodology must clearly specify those responsible for results. | 0.844 |
| Methodology should be teachable | 0.844 |

However, it is important to note that these results leave out factors that, in a way, are of great importance for METUIGA methodology as a methodology that works in a dynamic user-oriented environment. Moreover, it is important to indicate that within these six success factors are found three success factors identified by Méndez and Garrido [9] as fundamental desirable requirements in a software methodology. Likewise, assessing the METUIGA methodology evaluation matrix where the results of desirable requirements were weighted, we can conclude that the success factors are satisfied with an evaluation of 3.822 and 3.263, respectively in a range of 1 to 4. Finally, it is important when mentioning the results obtained that when evaluating if working in a dynamic user-oriented environment

with a score of 4 and 3, respectively, due to user-centered design approach that follows METUIGA methodology, these scores make us deduce that end users are being correctly included in each of METUIGA methodology stages.

## 9. Conclusions

"Tangible user interfaces have become a new field for innovation and entrepreneurship" [60]. This paper showed the METUIGA methodology development process, which aims to develop applications that use tangible interfaces and gamification techniques for blind people, a review of its stages and activities oriented to a correct implementation of gamification and tangible user interfaces, different methods to build tangible interfaces were demonstrated, as well as a review of gamification and tangible interface process implemented in requirements and design stages.

Currently, there are multiple works in the literature that implement the use of tangible interfaces focused on the education of children [61]. However, there are very few that consider educational topics for blind people using tangible user interfaces and gamification techniques. For this reason, the prototypes presented here are unique in mixing both the features following the development process proposed by the METUIGA methodology. Furthermore, all this research reports considerable advantages when improving learning in blind people so by combining both features can exponentiate learning and understanding in these specific users [15,16,20,21,62].

The first prototypes generated by the methodology were presented, which helped to improve the development process and demonstrated the capability to generate applications with tangible user interfaces and gamification features. In addition, the options for integrating the gamified elements with the tangible objects through challenges and missions were presented. Finally, a prototype oriented to children with ASD was presented, demonstrating how the METUIGA methodology can be used in different areas of interaction. The representative images of the interaction with the tangible interface were also shown.

The METUIGA methodology analysis was performed following the methodological evaluation model proposed by Méndez and Garrido [9]. The weighting of success factors was carried out with nine experts in software engineering, indicating that a methodology should cover the entire development cycle, integrate different development phases and be able to work in different software environments. These factors were identified as the three most important factors for a software methodology.

Based on the analysis of the results, it was concluded that the evaluators' perceptions indicate that the METUIGA methodology meets the desirable requirements of a software methodology according to the model outlined by the authors in [9] with a score of 3.822 and 3.263 in a range of 1 to 4. Additionally, we must consider that these results are limited to observations of the behavior of the METUIGA methodology. The next activity will involve more software developers. "It is possible to improve the effectiveness of methods significantly by involving multiple evaluators" [63].

These results motivate us to continue making functional prototypes to improve each of the stages and activities of the methodology to help build more applications focused on blind, visually impaired or otherwise disabled people and finally apply new evaluation methods to the METUIGA methodology to continue discovering aspects in which it can be improved. Additionally, we intend to analyze how children learn using their hands, explore the benefits of embodied metaphors in hybrid physical digital environments, and compare the use of tangible and graphical objects with the prototypes presented here [64–66].

**Author Contributions:** All authors have contributed equally to this work. All authors have read and agreed to the published version of the manuscript.

**Funding:** This research received no external funding.

**Institutional Review Board Statement:** Did not involve humans or animals.

**Informed Consent Statement:** Not applicable.

**Data Availability Statement:** In case you want to observe how these prototypes work, you can go to the following link: https://cutt.ly/anYHCAQ (accessed on 25 May 2021). In addition, to allow easy access to the third prototype, adaptations were made for its operation through tactile devices without the need to use tangible objects. Currently there are more than 500 downloads around the world. You can download this application at the following link for educational purposes: https://cutt.ly/ggELd4D (accessed on 25 May 2021).

**Acknowledgments:** We would like to acknowledge Engineers Uriel de Jesús Carbajal Morales and Leonardo Wenceslao Equihua Rodriguéz for their contribution to implement the assessment instruments for METUIGA methodology.

**Conflicts of Interest:** The authors declare no conflict of interest.

## Appendix A

### Challenges and missions to implement in the gamified system

*This section will define the different challenges and missions that will be implemented in the system. The challenges and missions that users can do must be in tune with the objectives of the client who asks to gamify his system.*

**System Challenges and Missions: Tangible Emotions**

SYSTEM CHALLENGES

| Title | Description |
| --- | --- |
| Happiness emotion | Identifies the emotion happiness within the system. |
| Sadness emotion | Identifies the emotion sadness within the system. |
| Fear emotion | Identifies the emotion Fear within the system. |
| Anger emotion | Identifies the emotion Anger within the system. |
| Surprise emotion | Identifies the emotion Surprise within the system |
| Enthusiastic Learner | User visits learning section |
| Explorer | User visits all system interfaces |

SYSTEM MISSIONS

| Title | Description |
| --- | --- |
| Apprentice | The user enters the system for the first time |
| Professional | The user has completed 75% of activities |
| Master | The user has completed 100% of the activities |
| Expert | The user has completed 100% of activities with the maximum number of stars |
| Connoisseur | The user identifies all the emotions of the system |
| Passes an activity | The user successfully completes an activity within the system. |

**Figure A1.** Challenges and missions established at design stage for the implementation of tangible emotions.

**Designing gamification structure of the system**

**Project gamification structure: Tangible Geometry**

*We decided to use only four stages of the gamification process for this project.*

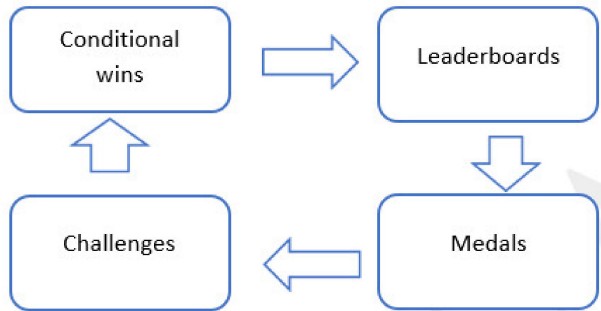

**Reward system: PBL**

*It was decided to use the PBL system which will be based on:*

*Point system*

- *Experience points: To be able to observe your progress in the activities.*

*Badges*

- *Offer feedback: We will offer feedback when you get a medal.*
- *Incentivize challenges and missions: At the end of an activity if it is successful, you will be awarded stars according to your performance.*
- *Encourage collecting: You will have a place to see the medals received.*

**Figure A2.** Gamification framework selected for the Geotag application at the time known as Geometria tangible.

**Dynamic interface designs involving end users**

**Dynamic design: Initial screen**

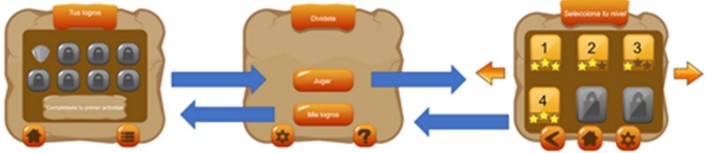

**Dynamic design: Activity display**

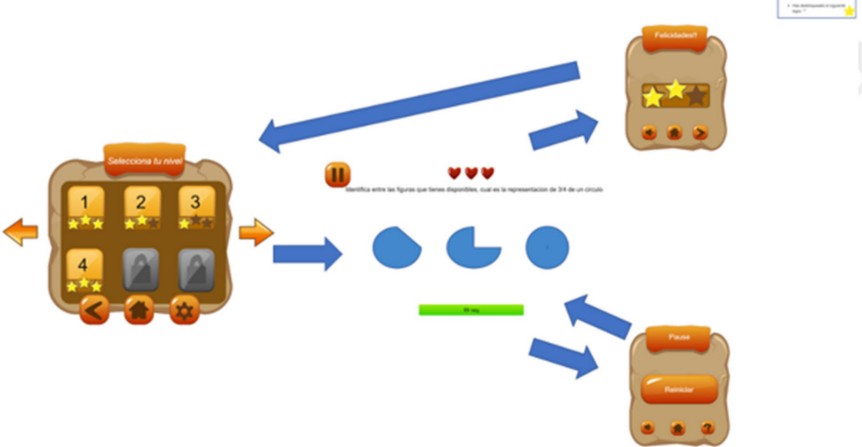

**Figure A3.** Initial designs of tangible portions application considering end users.

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
