# Peer review of "Methodology for Designing Systems Based on Tangible User Interfaces and Gamification Techniques for Blind People"

_applsci, doi:10.3390/app11125676_

Round 1

Reviewer 1 Report

Fhe article proposes a method and systems which are for children with Visual Disabilities and Autism (ADS).

- A) The article has serious problems with the background and all the previous work made on the field of tangible user interfaces.

Apart from the references [15, 16] the authors did not tell us that there is a lot of work on the field. For example see the titles:

1) Torino: A Tangible Programming Language Inclusive of Children with Visual Disabilities

2) Tangible User Interfaces to Ease the Learning Process of Visually-Impaired Children

3) Do Tangible User Interfaces promote social behaviour during free play? A comparison of autistic and typically-developing children playing with passive and digital construction toys

4) POMA: A tangible user interface to improve social and cognitive skills of Sri Lankan children with ASD  

5) Using tangible user interfaces in computer-based training systems for low-functioning autistic children

There is also work on the Design consideration of such tools. Please see the titles:

1) Evaluating children performance with graphical and tangible robot programming tools

2) Designing tangibles for children: what designers need to know.

3) Using developmental theories to inform the design of technology for children

4) Tangible User Interfaces: Past, Present, and Future Directions

Needless to say that tangible user interfaces have been used in a lot of cases with impaired people (please see the tile: Exploring the potential of programming tasks to benefit patients with mild cognitive impairment) but again there is nothing about it.

The article mention user evaluation and proposes tools for it. But again there is no information about what the others do to make user evaluation in the field.

Please see the titles:  

1) The evolving field of tangible interaction for children: the challenge of empirical validation

2) Tangible and graphical programming with experienced children: A mixed methods analysis

3) What the body knows: Exploring the benefits of embodied metaphors in hybrid physical digital environments

4) Exploring how children use their hands to think: An embodied interactional analysis

5) Comparing the use of tangible and graphical programming languages for informal science education

or what are the possible benefits might be. Please see the titles:

1) Latent Class Modeling of Children’s Preference Profiles on Tangible and Graphical Robot Programming

2) Are tangibles more fun? Comparing children's enjoyment and engagement using physical, graphical and tangible user interfaces

3) Using ‘tangibles’ to promote novel forms of playful learning

Please also you really need to explicitly say a few words about the systems that we can find out there and what is the unique thing that your ideas have to offer. A starting point to answer are the titles:

1) Tangible user interfaces for programming and education: A new field for innovation and entrepreneurship

2) Tangible interfaces in early years’ education: a systematic review

- B) Also you need to make the text more readable by reducing the length of the sentences. There very long sentences in many places

- C) The evaluation of nine experts, with no user evaluation, is a good reason to be discussed for the possible limitations    

 I hope that the above information will be helpful

Author Response

Dear reviewer,

We appreciate your time spent reviewing our paper.

Your comments have helped us to improve this paper, below is a table where you can see how your suggestions were addressed, also, I attach the article with these modifications. 

The article proposes a method and systems which are for children with Visual Disabilities and Autism (ADS).

- A) The article has serious problems with the background and all the previous work made on the field of tangible user interfaces.

Apart from the references [15, 16] the authors did not tell us that there is a lot of work on the field. For example see the titles:

·        Torino: A Tangible Programming Language Inclusive of Children with Visual Disabilities

·        Tangible User Interfaces to Ease the Learning Process of Visually-Impaired Children

·        Do Tangible User Interfaces promote social behaviour during free play? A comparison of autistic and typically-developing children playing with passive and digital construction toys

·        POMA: A tangible user interface to improve social and cognitive skills of Sri Lankan children with ASD 

·        Using tangible user interfaces in computer-based training systems for low-functioning autistic children

There is also work on the Design consideration of such tools. Please see the titles:

·        Evaluating children performance with graphical and tangible robot programming tools

·        Designing tangibles for children: what designers need to know.

·        Using developmental theories to inform the design of technology for children

·        Tangible User Interfaces: Past, Present, and Future Directions

Needless to say that tangible user interfaces have been used in a lot of cases with impaired people (please see the tile: Exploring the potential of programming tasks to benefit patients with mild cognitive impairment) but again there is nothing about it.

The article mentions user evaluation and proposes tools for it. But again there is no information about what the others do to make user evaluation in the field.

Please see the titles: 

·        The evolving field of tangible interaction for children: the challenge of empirical validation

·        Tangible and graphical programming with experienced children: A mixed methods analysis

·        What the body knows: Exploring the benefits of embodied metaphors in hybrid physical digital environments

·        Exploring how children use their hands to think: An embodied interactional analysis

·        Comparing the use of tangible and graphical programming languages for informal science education

or what are the possible benefits might be. Please see the titles:

·        Latent Class Modeling of Children’s Preference Profiles on Tangible and Graphical Robot Programming

·        Are tangibles more fun? Comparing children's enjoyment and engagement using physical, graphical and tangible user interfaces

·        Using ‘tangibles’ to promote novel forms of playful learning

Please also you really need to explicitly say a few words about the systems that we can find out there and what is the unique thing that your ideas have to offer. A starting point to answer are the titles:

·        Tangible user interfaces for programming and education: A new field for innovation and entrepreneurship

·        Tangible interfaces in early years’ education: a systematic review

-         B) Also you need to make the text more readable by reducing the length of the sentences. There very long sentences in many places

-         C) The evaluation of nine experts, with no user evaluation, is a good reason to be discussed for the possible limitations   

  I hope that the above information will be helpful

A)      The suggestions were taken and references mentioned were added to strengthen the work (Line 156-168), in addition, the works by K. Sitdhisanguan et al. , Antle et al. , Wyeth et al. , Sapounidis et a. were incorporated in Table 3 to strengthen the projects and offer more tools to identify characteristics of tangible user interfaces.

      *It is important to mention that these works are suggestions to identify tools when using tangible user interfaces or to learn about them.

B)      In evaluation section the following line was added: "EIt is important to mention that although these instruments are proposed by ME-TUIGA methodology, development team is free to use other types of tools to evaluate the developed systems" later the suggestions were taken and the mentioned references were added where we identified evaluation instruments that can be applied to such systems.  (line 303-319).

C)      Emphasis was made in the conclusions section on the existence of multiple works that implement tangible interfaces, however, very few are considered for blind people and mix characteristics of tangible interfaces with gamification techniques (line 593-595).

D)       The presented sentences were organized for better writing.

Best regards
M. Sc. Roberto Ramos Aguiar

Reviewer 2 Report

See attachment.

Author Response

Dear reviewer,

We appreciate your time spent reviewing our paper.

Your comments have helped us to improve this paper, below is a table where you can see how your suggestions were addressed, also, I attach the article with these modifications. 

1.       In this manuscript, a METUIGA methodology development process is proposed, which aims at developing applications that use tangible interfaces and gamification techniques for blind people. In the title of the manuscript “Methodology for designing systems based on tangible user interfaces and gamification techniques,” there is no mention about the research target which is the blind people. Therefore, it is suggested that the title reflects this fact.

2.       The overall manuscript structure could be improved. For example, there are some duplications of section titles, such as those of Sections 4 and 7 which are both “Results.” It is suggested that the structure of this manuscript be improved by the IMRD structure (introduction, methodology, results, discussion and conclusions). The improved structure of the manuscript would help the reader understand the manuscript content more easily.

3.       Three prototypes of METUIGA methodology for educational applications are proposed. Prototypes one and two are used in the application of blind people, but prototype three is used for children with Autism Spectrum Disorder. Maybe, the third prototype could be removed so that the research subject can be more focused on blind people. Similarly, all applications presented in this manuscript are aimed at children, and so it is suggested that this be improved for the research goal of helping blind people.

4.       Tangible user interfacing and gamification are two different techniques for applications, and the combination of these two techniques for blind people’s education applications has a positive value. But this combination is not a pioneering work. Also, this research is only for children, which is applied to limited educational applications.

5.                       Some suggestions are as follows.

a.   The proposed METUIGA methodology in this manuscript can be applied to all blind people, not only for children.

b.   The application examples in this study can be applied to a wider range of applications, not just for the purpose of educational learning.

c.   More uses of the hearing sense can be used in the user interfacing because hearing is also a very important sense of blind people.

6.       This manuscript is aimed at blind people, and it is appropriate to have a detailed analysis of the usability for blind people in order for the reader to understand that the proposed METUIGA methodology is indeed applicable to blind people.

7.       In addition, some suggestions for improving the paper are as follows.

(1) Figures 3 and 4 are missing. Please reorder figure numbers in the manuscript.

(2) If the tables or figures are drawn by yourself, it does not have to make the note of "own creation."

(3) In line 366, figure 8(b) should be corrected to be figure 12(b).

(4) In line 378, figure 8(b) should be corrected to be figure 13(b).

(5) In line 487, section 7.1 should be corrected to be section 7.2.

(6) In line 500, section 7.1 should be corrected to be section 7.3.

1.       The title of the manuscript was modified to "Methodology for designing systems based on tangible user interfaces and gamification techniques for blind people" as suggested to make the objective of this research clearer.

2.       Section 4 was modified to METUIGA Methodology leaving the structure of the article as follows:

a) Introduction

b) Problem

c) Background

d) METUIGA Methodology

e) Developed applications

f) Evaluation of software development methodologies

g) Result

h) Discussion

i) Conclusions

3.       The idea of showing prototype three is with the intention of demonstrating that our methodological process could be adapted to other types of target users and for this reason this prototype is presented. Please note lines (413-417).

4.       Sections in the article have been corrected to include blind people in the use of the methodology. It is only stated in the prototypes that the target user is blind children.

5.       Suggestions made

a) It has been established in general that the methodology works for blind people, only in the prototypes it is established that they were directed to blind children.

b) Information was added on the use of a specific audio system and a voice system to specify the achievements and goals obtained (Line 386-391) (421-425).

6.       The construction of METUIGA methodology has been gradually improving its software engineering process over time, the purpose of this work is to demonstrate the current state of the methodology, the prototypes that have been created with it and the results of evaluation of its development process for this reason in this work we do not show usability results of the applications presented.

7.       The modifications indicated by the reviewer in figures 3,4 were made. The term "own creation" was eliminated, and the figures indicated were corrected.

Best regards
M. Sc. Roberto Ramos Aguiar

Reviewer 3 Report

Authors improved comments and remarks previously suggested.

Now background knowledge and methodology are clearer and more in line with results and conclusions.

Author Response

Dear reviewer,

We appreciate your positive comments to our paper. 

Modifications recommended by the other reviewers have been made, please find attached the updated document.

Best regards
M. Sc. Roberto Ramos Aguiar

Reviewer 4 Report

The article would be more readable and accessible if the prototypes were used to illustrate the METUIGA process. It would then become clearer how its principles guided the design process of these applications. The way the paper is written now, the reader has to infer this a lot. 

As a social scientist and educator I always put a lot of emphasis on how the "story" is told. And I am convinced that the story would become more interesting and catching if the prototypes were introduced earlier.

Many sentences are very long and very often the articles are missing (mostly "the"). This also impacts heavily on the readability of the article.

Author Response

Dear reviewer,

We appreciate your time spent reviewing our paper.

Your comments have helped us to improve this paper, below is a table where you can see how your suggestions were addressed, also, I attach the article with these modifications.

1.      The article would be more readable and accessible if the prototypes were used to illustrate the METUIGA process. It would then become clearer how its principles guided the design process of these applications. The way the paper is written now, the reader has to infer this a lot. 

2.      As a social scientist and educator, I always put a lot of emphasis on how the "story" is told. And I am convinced that the story would become more interesting and catching if the prototypes were introduced earlier.

3.      Many sentences are very long and very often the articles are missing (mostly "the"). This also impacts heavily on the readability of the article

1. A section has been added in Appendix A where the instruments implemented in the construction of the prototypes can be observed (Line 641- 649), in addition, references have been added where tangible interfaces can be observed in operation, as well as a link to the TANGIBLE Emotions application modified to work on Android system (line 632 -365).

2. It has been decided to show a brief description of each prototype in the introduction section. (line 73-80).

3. An analysis of the article has been made to improve its readability as suggested.

Best regards

M. Sc. Roberto Ramos

Round 2

Reviewer 1 Report

Congratulations...

Regards,

Reviewer 2 Report

Most of the corrections have been completed.

This manuscript is a resubmission of an earlier submission. The following is a list of the peer review reports and author responses from that submission.

Round 1

Reviewer 1 Report

line 26, line 44, line 47, and so on - check punctuation

line 40 which methods? which advances? Provide examples

line 100 "we must fernandes and Healy"

No related researches given

line 124

How p. 3.1-3.2 are related to Results?

Figure 1 is not well explained. What arrows mean? What is the link between requirements, design, implementation, and evaluation?

From the text (pp. 3.3.1-3.3.3) is not clear what is the novelty of METUIGA methodology. 

line 221 - cite properly

I doubt that survey among 2 persons provides statistically significant results. What about the internal and external validity of the obtained results?

Overall, what is the novelty of the approach? You applied specific tools and techniques as far as your products have to be profiled to blind children. Why it should be called a new methodology?

Author Response

Dear reviewer,

we appreciate your comments to continue improving this work. 

Below you will find a table specifying how your comments were addressed, likewise I attach the updated document. 

Observation

¿How was it attended?

line 26, line 44, line 47, and so on - check punctuation

The general punctuation of document was reviewed as indicated

line 40 which methods? which advances? Provide examples

Examples of new technologies such as Augmented Reality, Virtual Reality, Tangible Interfaces were added. (Line 44-53)

line 100 "we must fernandes and Healy" 

The research given to fernandes and healty was modified. (Line 113)

No related researches given line 124

Added research related to tangible interfaces and blind people (146-153)

How p. 3.1-3.2 are related to Results?

Tangible user interfaces and gamification are part of background analyzed to create METUIGA methodology. (A new section was created to discuss these features and not add them into the results. New Section 3 Background)

Figure 1 is not well explained. What arrows mean? What is the link between requirements, design, implementation, and evaluation?

The function of arrows in figure 1 was explained. (Linea 204-207)

From the text (pp. 3.3.1-3.3.3) is not clear what is the novelty of METUIGA methodology.

Some of the most innovative activities of METUIGA methodology are explained in points (4.1-4.4) (All section 4).

line 221 - cite properly

Properly cited as indicated (Linea 295)

I doubt that survey among 2 persons provides statistically significant results. What about the internal and external validity of the obtained results?

We specified that it is a pre-analysis applied to two people from the Abstrac section (Line 18-21), finally, we specified in conclusions to involve more evaluators to obtain more significant data by referring to an article by NIELSEN (Line 474-476).

Overall, what is the novelty of the approach? You applied specific tools and techniques as far as your products have to be profiled to blind children. Why it should be called a new methodology?

Added a comparative table with other user-centered software methodologies (Line 210- 216) as well as a more detailed explanation of methodology novelties in its development stages (Section 4.1-4.4).

Best regards

M. Sc. Roberto Ramos 

Reviewer 2 Report

Topic is of great interest and actual; nevertheless, relevant issues have not been detected by authors:

  • background knowledge is not properly presented, limited to few referencing models and not taking into account the the prominent doctrine of Universal Design (UD) set by UE - the focus with disabilities and new technologies stays into the assumption that technology needs to be adapted to users and not the way around; more, technology needs to follow universal principles of design able to meet ALL users needs, without dividing audience between abilities vs. disabilities!
  • methodological approach is limited to software engineers, not taking into account the role of user experience designers, content developers, pedagogists, educators and cognitivists - TUIs are multidisciplinary products! 
  • Applications' description: the three prototypes are not fully described, especially the most interesting part of tangible objects and interfaces; maybe some images of the prototype can be useful to understand the interaction modes.
  • evaluation procedures: the number of evaluated users is really poor; probably tests cannot be rearranged...in this case, it's better to present the paper at the very beginning as a summative pre-analysis so to not induce readers to expect for massive evaluation. Make clear this point in the abstract as well!

Some English errors are present;  punctuation revision is needed too.

Author Response

Dear reviewer,

we appreciate your comments to continue improving this work. 
Below you will find a table specifying how your comments were addressed, likewise I attach the updated document. 

Observation

¿How was it attended?

background knowledge is not properly presented, limited to few referencing models and not taking into account the the prominent doctrine of Universal Design (UD) set by UE - the focus with disabilities and new technologies stays into the assumption that technology needs to be adapted to users and not the way around; more, technology needs to follow universal principles of design able to meet ALL users needs, without dividing audience between abilities vs. disabilities!

A background section was added, talking about tangible interfaces and gamification techniques and mentioning their advantages in teaching blind people. (Section 3)

methodological approach is limited to software engineers, not taking into account the role of user experience designers, content developers, pedagogists, educators and cognitivists - TUIs are multidisciplinary products! 

Added information related to the multidisciplinary work carried out in the construction of the applications. (Line274-287 )

Applications' description: the three prototypes are not fully described, especially the most interesting part of tangible objects and interfaces; maybe some images of the prototype can be useful to understand the interaction modes.

Representative images were added where you can see the tangible interface interaction and the prototypes presented. (Line 312,334,363)

evaluation procedures: the number of evaluated users is really poor; probably tests cannot be rearranged...in this case, it's better to present the paper at the very beginning as a summative pre-analysis so to not induce readers to expect for massive evaluation. Make clear this point in the abstract as well!

It was clarified that this is a pre-analysis applied to two people from the Abstrac section (Line 18-21), and finally, it was specified in the conclusions to involve more evaluators to obtain more significant data (Line 468-477).

Some English errors are present;  punctuation revision is needed too.

A linguistic revision of all documents was carried out.

Best regards

M. Sc. Roberto Ramos

Reviewer 3 Report

As mentioned in the title of the submitted manuscript, the authors deal with a methodological approach to the construction / designing systems. These systems include tangible user-interfaces and techniques of gamification. The paper has potential. However, there are a couple of points which require a thorough revision:

  • You introduce the rather general term “systems” in the introduction. To better classify the contents of your paper (and make it more interesting for the respective research community), you should add a describing term. What systems do you design based on tangible user-interfaces and gamification approaches?

  • The structure of your three chapters should be re-organized. I miss a clear chain of argumentation showing the readers, a) the research gap (why is the design of what systems – see point No.1 – important?, b) how are tangible user-interfaces addressed in related studies?, c) how are gamification techniques included in related studies?, d) why is there a lack of combining these two aspects in the way you are planning to do it? This logical chain should be followed by a method chapter describing your approach.

  • The gamification section is by far too short. I would like to read a solid introduction why gamification approaches have increased in the last couple of years. This has also to do with the free availability of game engines (from the mid-2010s onwards), which allow professionals and the broad interested public to develop new kinds of virtual environments. The increase of VR is closely related to gamification. The following references serve as examples: https://doi.org/10.3390/ijgi9110655 and https://doi.org/10.1007/s42489-019-00030-2

  • Could your conclusion chapter please include references to related studies? In the current version, the readers cannot get the information in how far your approach backs up, extends, contradicts etc. other related approaches. This would help to better position your study within the ongoing research debates

Author Response

Dear reviewer,

we appreciate your comments to continue improving this work. 
Below you will find a table specifying how your comments were addressed, likewise I attach the updated document. 

Observation

¿How was it attended?

You introduce the rather general term “systems” in the introduction. To better classify the contents of your paper (and make it more interesting for the respective research community), you should add a describing term. What systems do you design based on tangible user-interfaces and gamification approaches?

The term "technological systems" was changed to "educational technological systems" to give more emphasis to the systems developed (Line 66).

The structure of your three chapters should be re-organized. I miss a clear chain of argumentation showing the readers, a) the research gap (why is the design of what systems – see point No.1 – important?, b) how are tangible user-interfaces addressed in related studies?, c) how are gamification techniques included in related studies?, d) why is there a lack of combining these two aspects in the way you are planning to do it? This logical chain should be followed by a method chapter describing your approach.

The structure of chapter three was reorganized (chapter 3 of background and 4 of results were added). Studies related to Tangible Interfaces (Line 146-153) and Gamification (Line 174-182) were added. The intention of combining both technologies to take advantage of their benefits for blind people is explained (Line 186-190).

The gamification section is by far too short. I would like to read a solid introduction why gamification approaches have increased in the last couple of years. This has also to do with the free availability of game engines (from the mid-2010s onwards), which allow professionals and the broad interested public to develop new kinds of virtual environments. The increase of VR is closely related to gamification. The following references serve as examples: https://doi.org/10.3390/ijgi9110655 and https://doi.org/10.1007/s42489-019-00030-2

More background on gamification was shown as well as its explanation in the use of other technologies such as VR (Line 174-176) as well as previous work and advantages in blind people (Line 176-182).

Could your conclusion chapter please include references to related studies? In the current version, the readers cannot get the information in how far your approach backs up, extends, contradicts etc. other related approaches. This would help to better position your study within the ongoing research debates

Added references to previous papers that have worked with gamification and tangible interfaces with blind people (Line 455-461).

Best regards

M. Sc. Roberto Ramos 

Reviewer 4 Report

The authors shows prototypes to help in teaching geometry and mathematical portion in blind people, also focusing on children with autism spectrum disorder.

The proposed study is interesting but there are some points that the authors should better discuss.

The abstract should also include some findings of their analysis. The authors should be better described the novelties of their approach with respect to existing ones. The authors should provide more information about the survey and which users' population have responde to it. Furthermore, the authors should provide more details and discussion about the obtained results. The Discussion section also needs to be improved by analyzing the outcome of evaluation section.

I suggest to further analyze more recent approaches about the examined topics. In particular, I suggest the following papers to further investigate the relevance of emotional aspects and multimedia data for this topic in the intrduction section:

1) An emotional recommender system for music. IEEE Intelligent Systems.

2) Kira: a system for knowledge-based access to multimedia art collections. In 2017 IEEE 11th international conference on semantic computing (ICSC) (pp. 338-343). IEEE.

Finally, I suggest to perform a linguistic revision.

Author Response

Dear Reviewer,
we appreciate your comments to improve this work, below I show you a table of how your comments were addressed, in addition, I added the final version of updated document.

Observation

¿How was it attended?

The abstract should also include some findings of their analysis. The authors should be better described the novelties of their approach with respect to existing ones. The authors should provide more information about the survey and which users' population have responde to it. Furthermore, the authors should provide more details and discussion about the obtained results. The Discussion section also needs to be improved by analyzing the outcome of evaluation section.

Abstract was modified considering observations indicated, more information was added on the analysis and use of tangible interfaces and gamification (Line 8-21). The METUIGA methodology novelties are written in section 4.

I suggest to further analyze more recent approaches about the examined topics. In particular, I suggest the following papers to further investigate the relevance of emotional aspects and multimedia data for this topic in the intrduction section: 1) An emotional recommender system for music. IEEE Intelligent Systems. 2) Kira: a system for knowledge-based access to multimedia art collections. In 2017 IEEE 11th international conference on semantic computing (ICSC) (pp. 338-343). IEEE.

The recommended articles were analyzed and recommendation systems were added as one of the new methods to help in the interaction with users (Line 44-53).

Finally, I suggest to perform a linguistic revision.

A linguistic revision of all documents was carried out.

Best regard 

M. Sc. Roberto Ramos 

Round 2

Reviewer 1 Report

Lines 44-50 - much shorter, no need for definitions

Line 113 "fernandes and Healy" - should be fixed

Lines 102-115 What is the idea behind these long quotes?

Figure 1 - fix the legend

Figures 7-10 - out of boundaries

You have specified that it is a pre-analysis - so, what are the limitations and validity of the approach?

Author Response

Dear reviewer, we appreciate your comments in this second round to continue improving the work presented here.

Below is a table that addresses your comments and clarifies some others.

Observation

As attended

Lines 44-50 - much shorter, no need for definitions

-Dear reviewer, these definitions were added because previous reviewers suggested mentioning different types of technologies due to the above lines (42-44) that read as follows: "Technologies have evolved over the years to achieve greater familiarity and interaction with users, gradually appearing different methods capable of improving their experience with technology presented to them"

Line 113 "fernandes and Healy" - should be fixed

-The line indicated in the surname Fernandes has been corrected.

Lines 102-115 What is the idea behind these long quotes?

-Text on lines 104 and 105 was modified to indicate what is behind user-centered design definitions.

Figure 1 - fix the legend

The legend of table 1 was fixed because it was repeated with table 2, the legend of figure 1 was also modified.

Figures 7-10 - out of boundaries

- Figures 7-10 are not out of bounds, in MDPI's template two ways of presenting figures can be observed on the document. Therefore, we decided to implement the second option to represent these images.

You have specified that it is a pre-analysis - so, what are the limitations and validity of the approach?

- It was added that because this is a two-person analysis, these results are limited to a first observation of METUIGA methodology. To further validate the approach, it is specified that next activity will be to involve more software developers in evaluating METUIGA methodology.(Line 468-472)

I also share with you this document which has had minimal modifications in response to your comments.

Best regards

M. Sc. Luis Roberto Ramos Aguiar

Reviewer 2 Report

Although authors have a different point of view on the subject of the paper, I appreciated the suggested integrations done.

Some minor spelling revisions are still needed but the overall work is fine.

Author Response

Dear reviewer,

we appreciate the feedback you provided to improve this work. I share with you the latest version where modifications were minimal.

Best regards

M. Sc. Luis Roberto Ramos Aguiar

Reviewer 3 Report

The authors provided a revised manuscript version which takes up the suggestions made in the first review round. The response letter is rather short. However, the changes made show that the authors have revised the paper thoroughly. The new chain of argumentation is definitely improved. In my opinion, the revised manuscript can be accepted for publication.

Author Response

(The authors gave the same response as above.)

Reviewer 4 Report

I think that the authors have addressed all my concerns.

Author Response

Dear reviewer,

we appreciate the feedback you provided to improve this work. I share with you the latest version where modifications were minimal.

Best regards

M. Sc. Luis Roberto Ramos Aguiar.
